# Study on Mass Erosion and Surface Temperature during High-Speed Penetration of Concrete by Projectile Considering Heat Conduction and Thermal Softening

**DOI:** 10.3390/ma16093604

**Published:** 2023-05-08

**Authors:** Kai Dong, Kun Jiang, Chunlei Jiang, Hao Wang, Ling Tao

**Affiliations:** School of Energy and Power Engineering, Nanjing University of Science and Technology, Nanjing 210094, China; dongkai@njust.edu.cn (K.D.);

**Keywords:** concrete, penetration, melting and cutting erosion, temperature, heat conduction, coupling numerical calculation

## Abstract

The mass erosion of the kinetic energy of projectiles penetrating concrete targets at high speed is an important reason for the reduction in penetration efficiency. The heat generation and heat conduction in the projectile are important parts of the theoretical calculation of mass loss. In this paper, theoretical models are established to calculate the mass erosion and heat conduction of projectile noses, including models of cutting, melting, the heat conduction of flash temperature, and the conversion of plastic work into heat. The friction cutting model is modified considering the heat softening of metal, and a model of non-adiabatic processes for the nose was established based on the heat conduction theory to calculate the surface temperature. The coupling numerical calculation of the erosion and heat conduction of the projectile nose shows that melting erosion is the main factor of mass loss at high-speed penetration, and the mass erosion ratio of melting and cutting is related to the initial velocity. Critical velocity without melting erosion and a constant ratio of melting and cutting erosion exists, and the critical velocities are closely related to the melting temperature. In the process of penetration, the thickness of the heat affected zone (HAZ) gradually increases, and the entire heat conduction zone (EHZ) is about 5~6 times the thickness of the HAZ.

## 1. Introduction

Underground protective fortifications are mostly built with high-strength concrete and are mainly used to resist the shock wave generated by the explosion and the penetration of kinetic energy projectiles. The research on kinetic energy projectiles penetrating concrete is mostly concentrated on the depth of penetration (DOP), while the change of projectile kinetic energy, the optimization of nose shape, and ballistic stability are perceived as important research aspects [1,2]. According to the different initial velocities of the projectile, the penetration process can be divided into a rigid region of penetration, a semi-fluid region of penetration, and a hydrodynamic region of penetration by studying the interaction between the projectile and the target [3]. With the increase in the initial velocity of penetration, it is found that the projectile has increasingly obvious mass erosion [4], which causes nose deformation and even disintegration of the projectile during the penetration process; consequently, the penetration efficiency and ballistic stability will be seriously affected [5].

Research on mass erosion began in the 1990s. Forrestal and Frew [6,7] carried out a series of tests on the high-speed penetration of sharp projectiles into concrete and focused on the shape changes of the projectile before and after the penetration. Significant mass loss after the penetration of the projectile was observed, mainly manifested in the abrasion and deformation of the projectile nose, which led to a sharp reduction in the efficiency of improving DOP by increasing the projectile velocity. In the theoretical calculation of the DOP for the projectile penetrating concrete, the projectile is regarded as a rigid body with an unchanged mass in most simulation models. This assumption is reasonable when the initial impact velocity of the projectile is relatively lower [8,9]. However, the calculation models of rigid penetration under the conditions of medium or low initial velocity are not suitable for predicting the penetration with erosion at high initial velocity due to the disregard of mass loss. The linear relationship between the mass loss and the initial kinetic energy of the projectile was established through fitting the test data by Sliding et al. [10] and Chen et al. [11]. The calculation efficiency was greatly improved by the fitting model, but the applicability of the model was limited due to the lack of theoretical connotation.

In order to further understand the erosion mechanism and establish a scientific penetration-erosion theoretical model, Jones [12], He [13], and Guo [14] analyzed the metallography of the projectile after the test and found that there was a sign of metal melting and quenching in the heat affected zone (HAZ) of the nose surface. Adiabatic shear bands were also observed in some areas, indicating that the temperature of the nose was higher than the melting temperature during penetration. The molten metal layer would separate from the surface of the projectile and produce new molten liquid metal on the new layer. Obvious furrow scratches were also detected on the nose and fine aggregate particles were also embedded in the surface after the test, indicating that the surface of the nose was subjected to temperature softening and cut by the aggregate. Zhao [15] found, using the results of Forrestal’s test, that with the increase in aggregate hardness, the mass loss of the projectile became more obvious. In summary, the reasons for the mass reduction after penetration could mainly be attributed to the melting of the nose surface under high-speed friction and the cutting by the concrete aggregate at the interface. These findings provide a strong scientific basis for the mechanism and the numerical calculation of mass erosion.

Based on the erosion mechanism, He [16] and Li [17] successively carried out theoretical research on the model of erosion calculation and established a mass erosion calculation method combined with a high-speed friction theory and melting model. A coupling model of erosion calculation taking into account aggregate cutting and thermal melting was established by Ning [18], in which the accumulated friction energy was converted into thermal melting energy. Based on the assumption that the penetration is an adiabatic process, the mass loss is calculated by the receding of the nose surface. Guo [19] calculated the temperature within the thickness of the thin layer on the surface of the nose using the axial one-dimensional heat conduction model and obtained the temperature rise ratio of the typical position generated by friction heat.

There is a coupling relationship between the temperature change of the projectile and mass erosion. It is an important prerequisite for numerical calculation to further understand the energy-force-heat conversion mechanism of the projectile during the process of penetrating concrete. The softening, material flow, and mass loss of the projectile resulting from a high temperature and high-stress state are the mechanisms of material failure during the penetration process [14]. The essence of the projectile temperature rise is the conversion of plastic work to heat, and the heat generated by high-speed friction between the projectile and the target when the projectile impacts the concrete. The rapid rise of temperature can lead to the melting of the projectile surface, and the cutting efficiency was affected by the change of hardness of the HAZ on the projectile surface. On the other hand, when the temperature gradient is generated on the surface of the nose, part of the thermal energy will be propagated from the high-temperature zone to the low (including the interior of the nose and the concrete), and the conduction and dissipation of this part of energy should be considered when calculating the temperature and mass erosion of the nose.

Alloy steel with thermal softening behavior (including AISI 4340 alloy steel, et al.) is selected as the projectile material, and its strength and hardness will be reduced when exposed to higher temperatures [20,21]. The influence of thermal softening on friction cutting has not been discussed in existing theory. With further research, the temperature of the projectile in the process of penetrating concrete has gradually been paid attention to. It is unscientific to regard the nose surface as an adiabatic layer in the existing erosion model when calculating temperature. For the same projectile shape and concrete strength, the penetration duration is mainly determined by the initial velocity and the mass of the projectile. If the surface is taken as an adiabatic layer, the calculation will be less accurate with a longer penetration time; therefore, this consideration is key to scientifically predicting the erosion and temperature evolution law to carry out the coupling model with heat generation, heat conduction, and mass erosion of the projectile.

We conducted coupling calculations on the mass erosion and surface temperature during the process of projectiles penetrating into concrete. Based on the heat transmission theory, high-speed friction theory, and the conversion of plastic work into heat theory, the coupling model of projectile mass erosion and temperature on the nose surface in the process of penetrating concrete are investigated in this paper. In Section 2, the hardness caused by the temperature rise on the surface of the nose is described as dimensionless by the material constitutive model considering the temperature, and the cutting mass loss model during penetration is modified. The projectile temperature rise caused by flash temperature heat conduction and conversion of plastic work into heat is also considered. Models of coupling penetration, mass loss, and heat conduction were established. The erosion calculation of the penetration process was achieved through the assumptions and calculation process provided in Section 3. The numerical calculation of the dynamic mechanical parameters, surface temperature, and mass loss during the projectile penetration into concrete was carried out in Section 4. The effectiveness of the coupling erosion model was verified by comparing the calculation results and tests, while the mass erosion and surface temperature of the projectile were also calculated and analyzed.

## 2. Thermal and Dynamic Model of Penetration Process

Due to the interaction between the projectile nose and target during the process of penetrating concrete, the plastic deformation caused by extrusion and high-speed friction in the relative sliding process mainly appears at the nose of the projectile. Some researchers have shown that a small number of scratches were observed on the body, but they are negligible compared with the mass loss of the nose. Therefore, the mass erosion calculation always ignores the body. Compared with a medium and low initial velocity, the mechanical essence is more complex when the projectile penetrates concrete at high velocity. In particular, the conversion of work to heat has an apparent influence on the deceleration and temperature of the projectile in the penetration process. Thus, the calculation should couple the thermal and dynamic mechanical processes of the projectile in high-speed penetration.

In this paper, we summarize the research results of scholars, including their calculation models, which mainly consist of the dynamic mechanical model of the penetration, the conversion of plastic work into heat model, the temperature rise model, and the conduction of heat model of the projectile. These models are further researched in the next chapter.

### 2.1. Dynamic Mechanical Model in Penetration Process

The dynamic mechanical parameters of the penetration process have been studied in detail. The theoretical aspect is mostly based on the cavity expansion theory. The cavity expansion theory used for the calculation of the penetration process was originally proposed by Forrestal and Tzou [22] in 1997. The surface pressure of the projectile nose can be calculated by this theory, based on which the resistance function of the projectile can be described as well. The positive pressure on the surface of the projectile nose can be expressed as:(1)$$\sigma_{n}=Sf_{c}+\rho_{c}v_{n}^{2}$$
where *v*_*n*_ is the cavity expansion velocity of the target at the interface of the nose and target during penetration, which can be expressed as *v*_*n*_ = *v*_*p*_cosφ; *v*_*p*_ is the projectile velocity; and φ is the included angle between the projectile axis and the normal direction of the nose surface, as shown in Figure 1. Here, *ρ*_*c*_ is the density of concrete target and *f*_*c*_ is the compressive strength without confining pressure of concrete. The parameter *S* = 82.6*f*_*c*_^−0.544^ is fitted by test results [7].

(a) Crater stage:

It is assumed that there is no mass loss in the crater stage of the mass erosion calculation model [18]. Because the accumulated plastic work during the crater stage is relatively little, the temperature rise of the projectile is not significant as well, so the temperature of the nose surface does not reach the metal melting temperature. In this paper, the dynamic relationship of this stage is coupled with mass erosion in the numerical calculation for more accuracy, and the depth of the crater stage can be expressed as:(2)$$H_{1}=k^{\prime }d$$
where $k^{\prime }$ is a parameter which can be expressed as $k^{\prime }=0.707+h_{0}/d$, in which *h*_0_ is the length of the nose and *d* is the diameter of the projectile.

When the penetration depth is $x\leq H_{1}$, the axial resistance of the projectile in the crater stage can be expressed as:(3)F=cx
where *c* is a constant. According to the theoretical formula proposed by Forrestal et al. [23], the velocity of the projectile at the end of the crater stage can be expressed as:(4)$$v_{1}{}^{2}=\frac{m_{0}v_{0}^{2}-\left(pd^{3}k^{\prime }/4\right)Sf_{c}}{m_{0}+\left(pd^{3}k^{\prime }/4\right)N^{*}r_{c}}$$
where $m_{0}$ and $v_{0}^{}$ are the initial mass and velocity of the projectile impacting the target, respectively. $N^{*}$ is the shape factor of the projectile nose, which can be expressed as $N^{*}=\left(8\psi -1\right)/24\psi^{2}$, where ψ is the initial caliber-radius-head(CRH) of the projectile nose, defined as the ratio of the nose radius to the projectile diameter. According to the dynamic cavity expansion theory, considering the integral effect of penetration resistance on the projectile nose, the shape factor can also be defined as [24]:(5)$$N^{*}=-\frac{8}{d^{2}}\int_{0}^{b}\frac{y{y^{\prime }}^{3}}{1+{y^{\prime }}^{2}}dx$$
where *y* = *y*(*x*) is the boundary function describing the shape of the nose.

(b) Tunnel stage:

When the penetration depth is $x>H_{1}$, the projectile enters the tunnel stage. The pressure on the nose surface is divided by area and the axial resistance on the projectile is [23]:(6)$$F=\pi d^{2}\left(Sf_{c}+N^{*}\rho_{c}v^{2}\right)/4$$

The basic parameters such as the relationship between deceleration, velocity, penetration depth, the pressure of the nose surface, and the time of projectile in the process of penetration can be calculated based on the above dynamic mechanical models. During high-speed penetration, the mass erosion at the nose of the projectile involves physical and mechanical parameters including projectile friction, melting, temperature rise, and heat conduction, which need to be calculated in combination with the thermodynamic theoretical model.

Through the conclusions obtained from current research results, the results were calculated with fewer errors when the generation of heat on the nose part was considered only and the body part of the projectile was ignored. Therefore, the mass erosion and heat conduction of the projectile nose during penetration can be divided into four areas, as shown in Figure 2, which are, respectively: the mass loss zone, the high-temperature heat affected zone (HAZ), the heat conduction zone, and the undisturbed zone. Each zone has a different calculation method due to its generation mechanism, and adjacent zones are continuously distributed in the surface space of the nose and have a close thermodynamic relationship. These models in different regions based on their generation mechanisms will be analyzed and established in this paper.

### 2.2. Temperature Rise of Projectile Caused by the Conversion of Plastic Work into Heat at the Boundary

The pressure on the nose of the projectile is extremely high during the penetration, which causes elastic-plastic deformation of the surface. Moreover, the instantaneous dislocation of the lattice will produce an increase in internal energy, resulting in a temperature rise that is distributed in a gradient from the surface of the projectile to its interior part. The stress distribution of the projectile nose can be calculated iteratively using the surface pressure given by Equation (1).

The conversion of plastic work into heat is most evident in the HAZ. According to hardness analysis of the surface of the projectile after penetration by Jerome et al. [25], it has been found that the hardness in the HAZ region is relatively higher. This phenomenon indicates that the hardening process occurs after quenching when the projectile material is heated to the austenite transformation temperature. Therefore, on the basis of scientific evidence, we demarcate the temperature index of the HAZ inner layer as the boundary condition. In addition, the temperature (*T*_H_) range of the normal austenitizing of AISI-4340 steel for direct quenching is 1088K~1118K.

Molinari et al. [26] proposed a formula to calculate the HAZ thickness, which can be expressed as Equation (7), where $t_{p}$ is the time of heat conduction and the HAZ thickness, H(t), is the heat diffusion length in the characteristic time:(7)$$H\left(t\right)=\frac{1}{\varsigma }{\left(\frac{\lambda_{p}}{\rho_{p}c_{p}}\frac{L_{p}}{v_{r}}\right)}^{1/2}$$
Here, $\varsigma =1/\sqrt{2\pi }$ is the characteristic constant, $c_{p}$ is the specific heat capacity, $\rho_{p}$ is the density of projectile material, *L*_*p*_ is the length of the projectile, and *v*_*r*_ is the relative sliding velocity of the projectile and target. For 4340 steel, the thermal conductivity is $\lambda_{p}=44.5\;W/\left(m\cdot K\right)$.

In the process of penetration, severe plastic deformation occurs on the contact surface of the concrete, which consumes a lot of energy and results in a temperature rise. As one of the most widely used and classic thermodynamic constitutive models of metal materials at high strain rates, the Johnson-Cook (J-C) constitutive model composed of three polynomials [27] is applied to the calculation of projectile stress when penetrating the concrete. The strain rate term and temperature term are combined with the traditional stress-strain expression relationship. Thus, this model is applicable for the calculation of high strain rates and high temperatures in the penetration process. The flow stress, σ, can be expressed as a function of the effective plastic strain, ε, the effective strain rate, $\dot{\varepsilon }$, and the temperature, *T*. The J-C model is expressed as:(8)$$\sigma =\left(A+B\varepsilon^{n}\right)\left(1+Cln\frac{\dot{\varepsilon }}{{\dot{\varepsilon }}_{0}}\right)\left[1-{\left(\frac{T-T_{0}}{T_{m}-T_{0}}\right)}^{m}\right]$$
where ${\dot{\varepsilon }}_{0}$ is the reference strain rate; *T*_0_ is the room temperature; *T_m_* is the melting temperature of the projectile material; and *A, B, C, n,* and *m* are material constants.

The numerical method for constitutive equation calculation using the integration rate is introduced as the algorithm of stress update. The stress, $\sigma_{ij}\left(t+\Delta t\right)$, at t+Δt can be obtained by integrating the stress rate:(9)$$\sigma_{ij}\left(t+\Delta t\right)=\sigma_{ij}\left(t\right)+{\dot{\sigma }}_{ij}dt$$

According to the normal stress of the projectile described in Equation (1) and the J-C constitutive model, the relationship between the temperature gradient and time in the HAZ can be calculated. Furthermore, elastic-plastic work is converted to heat energy in the extrusion process, and the temperature rise of the nose can be expressed as [28]:(10)$$T_{w}=\frac{\beta }{\rho_{p}c_{p}}\int \sigma d\varepsilon$$
where β is the coefficient of the conversion of plastic work into heat and $c_{p}$ is the specific heat of the projectile material.

In summary, the temperature rise caused by the conversion of plastic work into heat in the HAZ and inner zone can be calculated by Equations (8)–(10).

### 2.3. Friction Cutting Considering Thermal Softening of Material

It should be noted that the surface of the nose is a special kind of metal with high temperature and high stress resistance that directly contacts the aggregate during the penetration process. In the present cutting model, it is assumed that the hardness of the projectile and aggregate is constant [17,18]. Because the aggregate particles are constantly updated on the surface of the nose in the penetration process, this assumption mentioned above will cause rare errors. However, cutting and melting are interrelated. In other words, the hardness changes caused by the high temperature of the surface have an important influence on the cutting of the nose.

The relationship between temperature, stress, and strain can be established based on the J-C model. Agreements are obtained that the hardness of the nose surface and the aggregate in the concrete both affect cutting erosion. The hardness of steel is related to heat-treatment. A large number of tests have shown that the hardness and yield strength of steel can be regarded as a linear relationship, approximately. According to the test results shown in Figure 3, the relationship between the tensile strength, $\sigma_{b}$(MPa), of steel and Brinell hardness, *H*_*p*_, is expressed as follows [21,29,30]:(11)$$\sigma_{b}=K\times H_{p}+b$$
where *K* and *b* are the fitting parameters obtained from the tests. For alloy steel, *K* can be taken as 3.36 and *b* can be taken as 30.92.

With a tensile strength of 1080 MPa, the corresponding Brinell hardness is about 320 HBS; AISI-4340 alloy steel is often chosen as a material in the design of penetrating projectiles. The hardness decreases when the strength of the material reduces because of the high temperature. Therefore, the hardness value can be obtained through the strength value. Since the Mohs hardness of quartz aggregate has been adopted in the theoretical formula in literature [18] for calculation, the parameter of relative hardness is introduced in order to adapt to the general calculation of the model, and the dimensionless analysis of the softening behavior of AISI-4340 steel can be described. The softening relative hardness ratio of projectile material is defined as the ratio of the initial hardness under room temperature to the softening hardness under high temperature:(12)$$\eta \left(t\right)=\frac{H_{\mathrm{p-Normal}}}{H_{\mathrm{p-Soft}}\left(t\right)}$$

In the process of penetration, the hardness of concrete aggregate, $H_{c}$, is defined as a constant. In Equation (12), the hardness of alloy steel, $H_{\mathrm{p-Soft}}\left(t\right)$, is a variable parameter related to the penetration time. The initial hardness is the test result at laboratory temperature. The hardness of surface metal will decrease with the rise of temperature.

In the classical Rabinowicz cutting theory [31], using single-point abrasive particles to apply a load of *p* on a soft metal surface and press it into the depth of *h*, the volume expression of cutting when the wear sliding unit distance on the metal surface is:(13)$$V=K\frac{p}{H_{m}}$$
where $H_{m}$ is the Mohs hardness of the bearing surface and *K* is the wear coefficient which depends on the hardness of abrasive particles and the hardness of the metal.

Through microscopic inspection and observation on the surface of the recovered projectile after penetration, it is found that the furrow shape of the nose surface distribution is similar to the particle wear. Equation (13) can be borrowed in cutting calculation. For penetration, the pressure, *p*, of particles can apply the nose surface pressure described in Equation (1). Since the hardness of steel is proportional to the yield strength, the relative hardness described in Equation (12) can be introduced to modify the classical cutting model. Therefore, the change of hardness on the nose surface caused by melting is considered in the model, and the modified cutting volume of a unit area can be expressed as:(14)$$dV_{c}=\eta K_{1}\frac{\sigma_{n}}{Y}v_{r}dt$$
where *K*_1_ is the wear parameter applicable to the projectile penetrating concrete, which is related to the relative hardness of the contact. The specific calibration method has been studied in the literature [18]. *Y* is the yield strength of the projectile material and $v_{r}$ is the relative sliding velocity of the projectile target surface, which can be expressed as $v_{r}=v_{p}sin\varphi$ (see Figure 1). Therefore, for the mass loss caused by cutting, the cutting quality per unit area can be expressed as:(15)$$\Delta m_{c}=\rho_{p}\times \Delta V_{c}=\rho_{p}K_{1}\int_{0}^{t_{e}}\eta \frac{\sigma_{n}v_{r}}{Y}dt$$

The strength, temperature, and hardness of the projectile can be calculated, respectively, and the mass loss models of the projectile considering cutting and thermal softening during the penetration of concrete are established based on these equations above.

### 2.4. Melting and Heat Conduction

For the calculation of the temperature rise of the projectile nose, it is not scientific to assume the melting of the layer on the surface of the projectile is an adiabatic process. The temperature rise caused by the conversion of plastic work into heat is applied to the whole nose. The friction on the surface of the projectile will produce a “heat supply”. There will be heat conduction from the surface of the projectile to the interior when the “heat supply” capacity is higher than the temperature rise of the plastic work.

#### 2.4.1. The Heat Conduction of Flash Temperature

The friction under high-speed sliding is very different from that under low-speed sliding on the surface. During high-speed penetration, the surface temperature will reach and even exceed the melting temperature [32,33]. Meanwhile, the large deformation, phase transformation, and melting at the friction interface make the analysis more complex in penetration. The friction heat generated between the projectile and the target during penetration is also a complex physical problem to solve. Moreover, the heat transfer rate involved in the calculation of heat conduction is a time-dependent parameter. The heat energy is concentrated on the thin-layer surface area of the projectile nose, and the temperature rises in a very short time, in what is known as “flash temperature”.

During high-speed penetration, the temperature of the nose surface rises rapidly due to the intense heat flow of the surface melting, resulting in the heat conduction to the internal projectile, to be specific, which is called the heat conduction of flash temperature. In the flash temperature stage, the surface temperature of the nose is much higher than that of the interior. Although the metal melting only takes a very short time and can be considered to be completed instantaneously, the continuous high-temperature liquid melting material has a great influence on the temperature of the internal projectile, especially after the temperature of the exposed surface layer has changed and affected the subsequent penetration parameters.

#### 2.4.2. Melting Temperature and Mass Loss

The molten metal on the surface of the projectile is separate from the projectile. The key to the calculation is the temperature of the liquid metal layer covering the surface of the nose. In this paper, the hypothesis that there is no temperature gradient in the liquid metal is proposed. If the conversion of plastic work into heat in the melting zone is ignored, it can be assumed that all the heat comes from friction. The heat flux density in the melting zone can be calculated according to the following equation:(16)$$q=f\cdot v_{r}=q_{p}+q_{c}$$
where $q_{p}$ is the heat flowing to the projectile nose per area and $q_{c}$ is the heat flowing to the concrete target per area; $q_{p}$ and $q_{c}$ can be expressed as:(17)$$q_{p}=-\lambda_{p}\frac{\partial T}{\partial {\vec{n}}_{p}},q_{c}=-\lambda_{c}\frac{\partial T}{\partial {\vec{n}}_{c}}$$
where ${\vec{n}}_{p}$ and ${\vec{n}}_{c}$ are defined as the normal direction along the relative sliding surface of projectile and concrete, respectively.

The friction is related to the positive pressure, $\sigma_{n}$, on the surface of the projectile nose. The coefficient of dynamic friction is taken as *μ* and the friction can be expressed as [34,35]:(18)$$f=\mu \sigma_{n}+\tau_{0}$$
where $\tau_{0}$ is the shear strength of concrete, and the relationship between the shear strength and the compressive strength without confining pressure is expressed as $\tau_{0}=f_{c}/\sqrt{3}$. The dynamic friction coefficient is the key parameter in Equation (18). Many tests have shown that the coefficient of friction is not a constant value, but varies when the projectile is at different relative sliding velocities. In this paper, the coefficient of friction proposed by Klepaczko is used in the model. The coefficient of friction takes into account adiabatic shear, thermal conductivity, and other factors, and is expressed as:(19)$$\mu =\frac{\tau_{0}}{p}\left(\frac{c^{\prime }}{\Lambda }\right)\left[\langle 1-{\left(\Theta_{b}\left(v\right)\right)}^{2}\rangle +\frac{\beta \left(\Theta_{b}\right)v}{\tau_{0}h}\right]\cdot \left[1-\left(1-f_{a0}\right)exp\left(-D\left(p-p_{0}\right)\right)\right]$$

The parameters in Equation (19) can be found in the literature [36]. As shown in Figure 4, the relationship between the coefficient of friction and the relative sliding velocity clearly indicates that the friction coefficient decreases with the increase in velocity.

Conventional projectiles are usually axisymmetric. The nose section area is divided into *N*_x_ and *N*_y_ points in the horizontal and vertical directions, so the number of grids is (*N*_x_ − 1) × (*N*_y_ − 1). Therefore, the temperature at any point can be calculated. The melting area of every time-step under two-dimensional conditions can be calculated based on the temperature. The melting volume can be obtained by integrating along the symmetrical axis. Then, the melting volume per unit time-step (*t*_*n*−1_ to *t*_*n*_) can be expressed as:(20)$$\Delta V\left(T,t\right)=V\left(T,t_{n}\right)-V\left(T,t_{n-1}\right)$$

Therefore, the temperature rise of the projectile nose during penetration can be calculated by Equation (10), while the loss mass of melting per unit area of the nose can be calculated by Equation (21) from the initial time to the end time of penetration, *t*_*e*_:(21)$$\Delta m_{m}=\rho_{p}\times \Delta V_{m}=\int_{0}^{t_{e}}\Delta V\left(T,t\right)dt$$

#### 2.4.3. Calculation of Temperature Distribution of Projectile Nose during Penetration

According to Section 2.2, the calculation of the temperature distribution of the projectile nose includes the flash temperature rise caused by friction, the temperature rise caused by the conversion of plastic work into heat, and the heat conduction of the nose. Based on Fourier’s law and heat balance theory, the basic theory for the heat conduction of the projectile follows Equation (22):(22)$$\rho_{p}c_{p}\frac{\partial T_{t}}{\partial t}=\lambda_{p}\left(\frac{\partial^{2}T_{t}}{\partial x^{2}}+\frac{\partial^{2}T_{t}}{\partial y^{2}}\right)$$

The initial condition for solving Equation (22) is $T\left(x,y,0\right)=T_{0}=298\;K$; the boundary condition of heat flow flash temperature is $q_{p}\left(x,y,t\right)=-\lambda_{p}\frac{\partial T}{\partial \vec{n}}$; the boundary temperature of the inner layer in HAZ is *T*_H_ = 1088 K~1118 K.

It can be seen from Equation (16) that the heat flow boundary is closely related to relative velocity, and the flash temperature period mainly exists in the high-speed stage of penetration. Melting and cutting erosion may have different proportions due to different speed stages, which are introduced in detail below.

## 3. Calculation Algorithm of The Coupling Model

### 3.1. Assumptions

Since the penetration process is an extremely complex mechanical process, the numerical calculation needs to be carried out with some assumptions, as follows:The projectile penetrates concrete normally, and the projectile is regarded as a standard axially symmetrical structure;The mass loss and conduction of heat during the penetration process only occurs at the projectile nose;The material of the projectile is isotropic, its density and thermal conductivity remain stable during penetration;The influence of phase changes of material on penetration and heat conduction is ignored;The aggregate particles in concrete are evenly distributed.

### 3.2. Discretization

With a thickness scope of the HAZ from microns to millimeters, a multi-scale discretization method is adopted in order to improve the calculation efficiency. As shown in Figure 5, the micro-scale grid division is employed on the surface of the projectile and the independent verification of the size grid is conducted during the calculation.

### 3.3. Coupled Algorithm: Simulation of Penetration Process

When the assumptions described in Section 3.1 are determined, the thermal-mechanical-erosion of the projectile penetrating concrete can be calculated based on the model established in Section 2. The calculation process is as follows:Inputting the initial parameters and initial boundary conditions of the projectile and target;Discretization of the projectile nose;Calculating the mechanical parameters in the dynamic process of the projectile in the crater stage and tunnel stage;Calculating the surface pressure and stress of the projectile nose;Calculating the flash temperature boundary, conduction of heat, plastic work, and total temperature of the nose surface.Updating the stress state and the relative hardness of the surface;Calculating the cutting mass loss and the melting mass loss;Updating the nose shape and the mass of the projectile;Increasing the time-step and repeating calculations from step (3) till the projectile velocity drops to zero.

## 4. Results and Discussion

### 4.1. Comparison between Theoretical Calculation and Tests

#### 4.1.1. Theoretical Prediction and Experimental Results of Mass Erosion

In the theoretical calculation of the DOP prediction, the most representative formula that calculates the depth of penetration when projectiles penetrate semi-infinite targets was proposed by Forrestal [37], which is a semi-empirical formula based on the test results, and is expressed as follows:(23)$$H=\frac{2m_{0}}{\pi d^{2}\rho N}ln\left(1+\frac{\rho Nv_{1}^{2}}{Sf_{c}}\right)+2d$$

According to the test results, Silling points out that when the penetration velocity does not exceed 1000 m/s [10], the mass loss of the projectile has a linear relation with the initial kinetic energy. The mass loss remains at a constant level while the velocity is over 1000 m/s, and the mass loss rate of the projectile can be described by Equation (24).
(24)$$\delta =\frac{\Delta m}{m_{0}}=\left\{ \begin{array}{l} c_{0}\cdot v_{0}^{2}/2\;,\;v_{0}\leq 1000\;m/s\\ \;c_{0}/2,\;v_{0}>1000\;m/s \end{array}\right.$$
Here, Δm is the mass loss after penetration and $c_{0}$ is the constant fitted according to the tests.

The molten metal separated from the surface of the projectile and the Mohs hardness of the aggregate in the target are taken into comprehensive consideration for the mass loss calculation of the projectile by He [38]. Ignoring the application limiting of the penetration velocity, the model proposed by Jones et al. [12] is modified as follows:(25)$$\Delta m=\eta_{a}\frac{\pi d^{2}\tau_{0}N_{1}^{*}H}{4\kappa Q}$$
where $\eta_{a}$ is the parameter related to Mohs hardness of the aggregate, $N_{1}^{*}$ is the initial value of the dimensionless longitudinal cross-sectional area of the nose, κ = 4.18 J/cal is the mechanical equivalent of heat, *Q* is the melting heat of the unit mass projectile material, and *H* is the ultimate DOP of the rigid projectile. Those models are representatives for predicting DOP and mass loss, which will be compared with the present model in the numerical calculation in the next section.

#### 4.1.2. Comparison and Analysis of Calculation Results with Test Results

In order to determine the rationality of the model considering the temperature to predict the mass loss of the projectile established in this paper, the test results in the literature [6,7] are used for comparison. The parameters and material properties of the projectile and target required for calculation in the test are shown in Table 1, Table 2, Table 3 and Table 4.

The consistency between test results and the theoretical prediction is the standard for verifying the effectiveness of theoretical predictions. Based on the parameters in Table 1, Table 2, Table 3 and Table 4, the DOP in Case 1–Case 4 obtained by using the model and algorithm established in this paper is shown in Figure 6. It can be seen that the calculation results of DOP by the model and calculation method considering the erosion effect established in this paper is in good agreement with the test results. The DOP of the projectile considering mass loss is lower than when mass loss is not considered. Because the parameter *S*, which affects the DOP, is obtained by result fitting, it is necessary to calibrate this value for the requirements of erosion penetration calculation.

The consistency for deceleration is found in the influence of mass erosion for the calculation of Case 1–Case 4, so we only take Case 1 to analyze in this paper. The deceleration was calculated and the results, whether considering mass erosion or not, are shown in Figure 7. In the process of projectile penetration, the mass of the projectile and the CRH of the nose decreases gradually, resulting in an increase in the deceleration. It can be seen that mass erosion is one of the most important factors affecting the penetration efficiency of the projectile. The mass loss is considered in the numerical calculation in the crater stage. Due to the influence of mass erosion, the deceleration in the crater stage is greater than that without mass erosion consideration, especially for low initial velocity conditions. With the increase in initial velocity, the influence of mass loss on the deceleration gradually decreases in the crater stage. This is mainly because the mass loss in the crater stage is a parameter related to time. The projectile with higher initial velocity has a shorter crater time and less melting mass loss for the same crater depth, so it has less influence on deceleration.

Due to the influence of mass erosion, the deceleration is higher than in the calculation data which does not consider the mass loss, especially in the middle and final stages of the whole penetration process, so the penetration time is shortened. With the increase in the initial penetration velocity, the cumulative mass erosion has a deeper influence on the deceleration in the tunnel stage.

#### 4.1.3. Analysis of the Proportion of Mass Loss for Two Mechanisms

The proportion of mass loss caused by cutting and melting in total mass erosion is different. In order to further reveal the mechanism of mass loss during high-speed penetration, quantitative analysis of the two mechanisms of mass erosion is convenient for the design and optimization of the projectile. The percentage of cutting erosion and melting erosion in the total mass loss is calculated and shown in Figure 8. It can be clearly seen that the mass loss caused by melting is relatively higher in the velocity scope of the study, indicating that the melting of the nose surface is the main factor of mass erosion. In Case 1 and Case 2, the mass loss caused by melting contributes to about 81% of the total mass loss. In Case 3 and Case 4, the mass loss caused by melting contributes to 91% of the total mass loss due to the lower hardness of the aggregate. It can be seen that the proportion of the mass loss caused by the two mechanisms is closely related to the hardness of the aggregate. When the projectile uses the same material, the higher the hardness of the aggregate, the higher the cutting erosion proportion. In the velocity range from 400 to 1200 m/s, the melting-cutting erosion ratio and velocity have low relevance, and the mass loss of the projectile is approximately linear with the initial velocity. In this view the linear fitting of Equation (24) in this velocity range is reasonable.

#### 4.1.4. Critical Velocity of Erosion Affected by Melting v_s_, v_c_

Case 3 had the condition that the projectile penetrates concrete at an initial velocity of 445 m/s. When the velocity decreases to about *v*_s_ = 370 m/s, the melting mass decreases rapidly and the inflection point appears. The surface of the nose will not melt when the velocity decreases to *v*_c_ = 106 m/s, and only cutting erosion occurs in the subsequent penetration process, as shown in Figure 9. When the projectile penetrates at an initial velocity of 1069 m/s, there is no disappearance of melting erosion even as the velocity decreases to the critical velocity. This phenomenon is related to the time accumulation of heat conduction of flash temperature. Because the velocity of heat conduction is higher than the receding velocity of erosion, the HAZ layer is a zone with a temperature gradient. When the projectile penetrates at low initial velocity, the influence of the initial melting flash temperature on the heat conduction is negligible, resulting in a small thickness of the HAZ layer. The surface internal energy is not enough to melt the material when the velocity is reduced to the critical velocity, *v*_c_, and only cutting erosion occurs after that. In the case of high-speed penetration, the HAZ surface temperature of the projectile surface is still high even though the projectile has reduced to the critical velocity, and the energy generated by friction is enough to heat the surface to the melting temperature.

#### 4.1.5. The Influence of the Melting of Mass Erosion

It can be inferred from Section 4.1.3 that when metal materials with high melting temperatures are used on the surface of the nose, mass erosion can be largely reduced. For high melting temperatures such as wolfram (melting temperature: 3708 K) and molybdenum (melting temperature: 2918 K), the final mass erosion at different initial penetration velocities is calculated and compared with 4340 steel, assuming that the hardness of the two materials is equal with that of steel. As shown in Figure 10, the material with a high melting temperature as the projectile surface can significantly reduce the mass loss in the process of penetration at the same initial penetration velocity.

It can be seen from the previous section that there is a critical velocity, *v*_c_, for losing the melting erosion. In order to explore the law of melting at different initial velocities, the mass erosion during penetration at different initial velocities is calculated. It is found that *v*_c_ increases with the increase in the melting temperature of materials.

When the velocity is lower than *v*_c_, the mass loss is caused by cutting completely. The surface temperature of the projectile nose is relatively low, especially at low speed, and does not reach the melting temperature, so the mass loss of the projectile is less due to the thermal softening behavior. When the initial velocity exceeds the *v*_c_, melting erosion emerges. With the increase in initial velocity, the proportion of melting erosion increases gradually. At critical velocity, *v*_s_, the ratio of mass loss caused by melting and cutting tends to be stable, and melting erosion is the main factor of mass loss. For Case 1 in this paper, the *v*_c_ of 4340 steel is 175 m/s and the *v*_s_ is 379 m/s; the *v*_c_ of molybdenum under the same hardness is 311 m/s and the *v*_s_ is 530 m/s; and the *v*_c_ of wolfram under the same hardness is 364 m/s and the *v*_s_ is 616 m/s.

The core reason for the reduction or disappearance of melting erosion is that the melting efficiency is reduced or even insufficient due to the reduction in heat on the projectile surface. The critical velocity, *v*_c_ and *v*_s_, is affected by multiple factors such as velocity, melting temperature of the material, nose shape, properties of concrete, and the temperature of the HAZ caused by heat conduction during penetration. Further quantitative analysis is not conducted in this paper, and detailed numerical calculation is required for specific conditions.

### 4.2. Temperature Field of the Nose Surface

Analyze the condition in which the projectile penetrates concrete at an initial velocity of 1069 m/s in Case 1. The temperature fields of the nose at different times during the penetration process are shown in Figure 11. The molten liquid metal with flash temperature (over 3000 K) is eliminated. Due to the extremely short time for the conduction of heat, the HAZ is only on the surface of the projectile and the surface temperature is between 1500 K to 1793 K, which is consistent with the law obtained by Guo [19]. However, the dynamic boundary effect of the erosion process is also considered in this paper, and the results are more scientific and accurate.

The austenitizing temperature for direct quenching of AISI-4340 steel is about *T*_H_ = 1088 K. It is defined as the HAZ where the surface temperature is higher than *T*_H_, and the heat conduction zone where the surface temperature is higher than *T*_0_ = 298 K. The thickness changes of the HAZ and the entire heat conduction zone (EHZ) on the nose surface at the nose tip (Point A), the nose middle (Point B), and the nose end (Point C) during the penetration process are shown in Figure 12. The result shows that the thickness of the HAZ increases with the increase in the penetration time, and the thickness of the EHZ is about 5~6 times of the thickness of HAZ. The thickness of the HAZ and EHZ at the tip is significantly higher than those at the middle and end of the nose and the thicknesses of the HAZ at the middle and end of the nose are approximately equal. The thickness of the heat conduction zone at the end of the nose is slightly higher than that at the middle. This distinction is mainly caused by the difference in erosion thickness. The erosion thickness gradually decreases from the tip to the end of the nose, and the receding displacement at the tip is significantly higher than that at the middle and end. The difference in HAZ thickness between the middle and the end of the nose is quite slight, but the difference in erosion thickness has an influence on the EHZ.

## 5. Conclusions

In this paper, a coupling calculation model of penetration, mass erosion, and heat conduction in the process of the projectile penetrating concrete at high speed is established, and the numerical calculation of dynamic mechanical parameters, surface temperature, and mass loss in the process of penetration is carried out. The effectiveness of the coupling erosion model is verified by comparison with experimental data, and the internal mechanism of the erosion process is analyzed. The maximal velocity of projectile for the erosion study is 1201 m/s in this paper and the projectile may be destroyed once it exceeds this limited velocity. The conclusions are as follows:

(1) When the projectile penetrates concrete at high speed, the influence of mass erosion on the deceleration of the projectile increases with the increase in initial velocity. The proportion of melting erosion is higher than that of cutting erosion. The proportion of cutting and melting is closely related to the hardness of the aggregate in concrete. The higher the hardness of the aggregate, the higher proportion of cutting mass loss.

(2) There are critical initial velocities of the projectile without melting erosion and critical initial velocities with stable proportional for melting and cutting erosion. The critical velocity is mainly related to the melting temperature of the material on the nose surface. When the velocity is lower than critical velocity *v*_c_, only cutting erosion occurs; when it is higher than *v*_c_ but lower than *v*_s_, the proportion of melting increases with the increase in velocity. When the velocity is higher than critical velocity *v*_s_, the ratio of melting and cutting erosion does not change with the different initial velocities.

(3) The thickness of the high-temperature heat affected zone and the entire heat conduction zone on the surface of the projectile nose are closely related to the time of penetration, and both increase with the increase in penetration time. The thickness of the entire heat conduction zone is about 5~6 times of the high-temperature heat affected zone.

## Figures and Tables

**Figure 1 materials-16-03604-f001:**
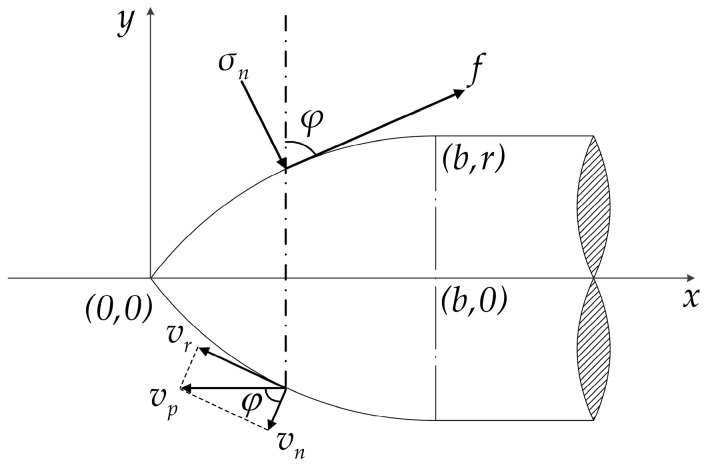
Two-dimensional sectional coordinates of projectile nose.

**Figure 2 materials-16-03604-f002:**
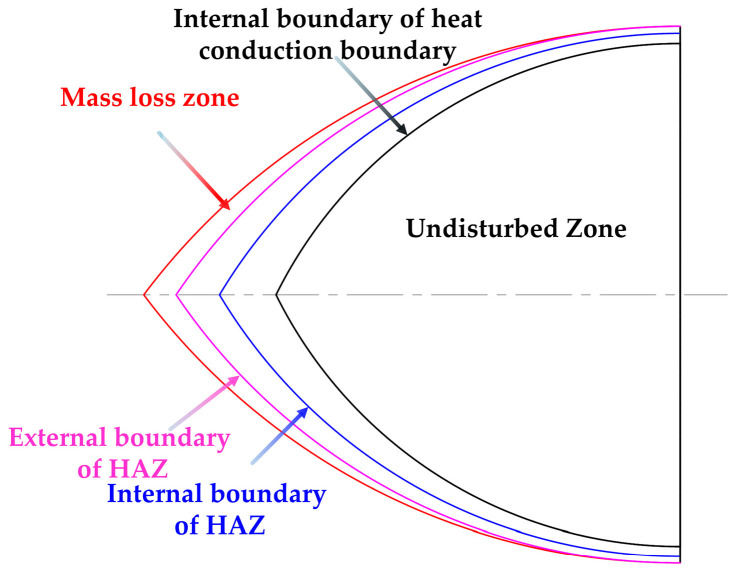
Thermal boundary of the projectile nose during penetration.

**Figure 3 materials-16-03604-f003:**
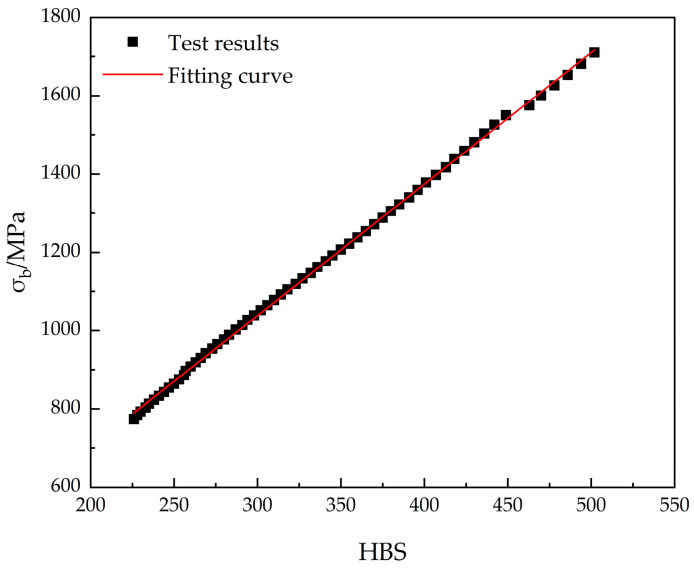
Relationship between Brinell hardness and tensile strength [29].

**Figure 4 materials-16-03604-f004:**
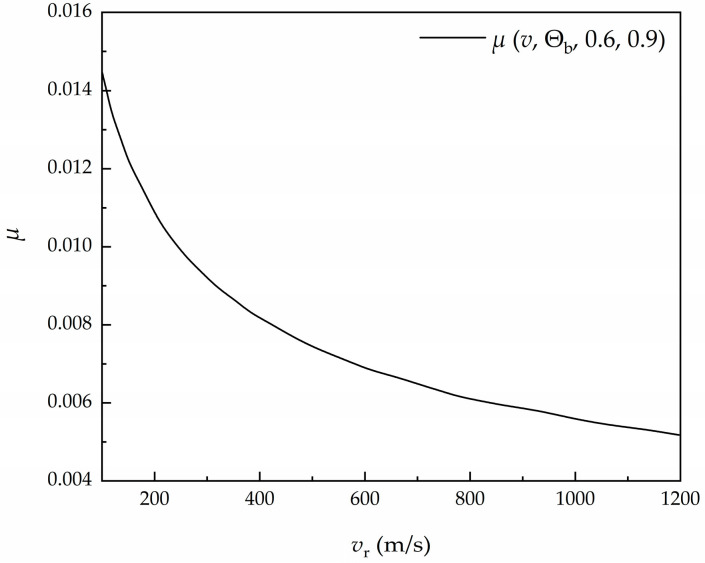
Relationship between coefficient of friction *μ* and sliding velocity *v*_*r*_ [36].

**Figure 5 materials-16-03604-f005:**
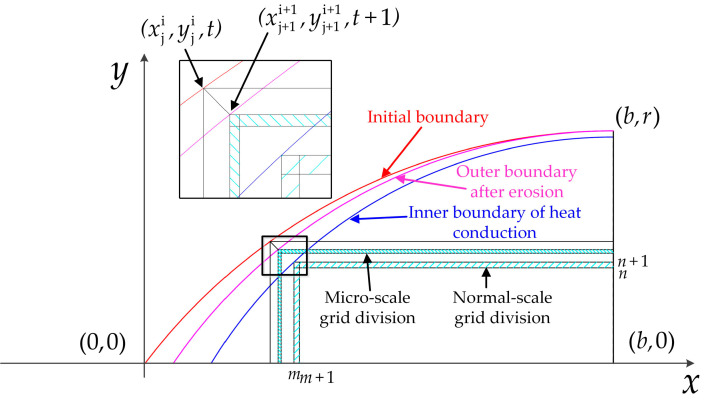
The micro-scale grid division of the receding model of the nose at time *t*.

**Figure 6 materials-16-03604-f006:**
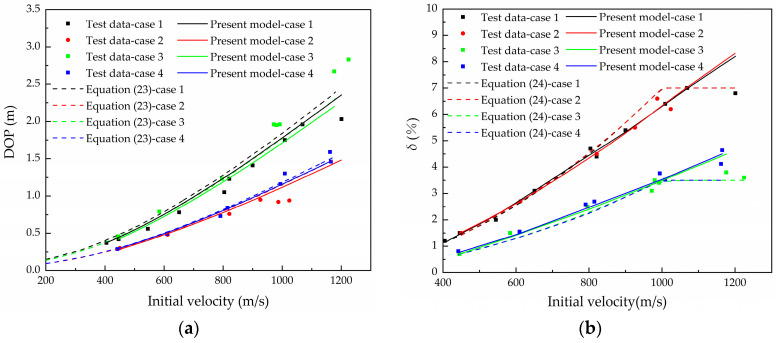
Comparing predicted results with different models and test data in Cases 1–4: (**a**) DOP, (**b**) *δ*.

**Figure 7 materials-16-03604-f007:**
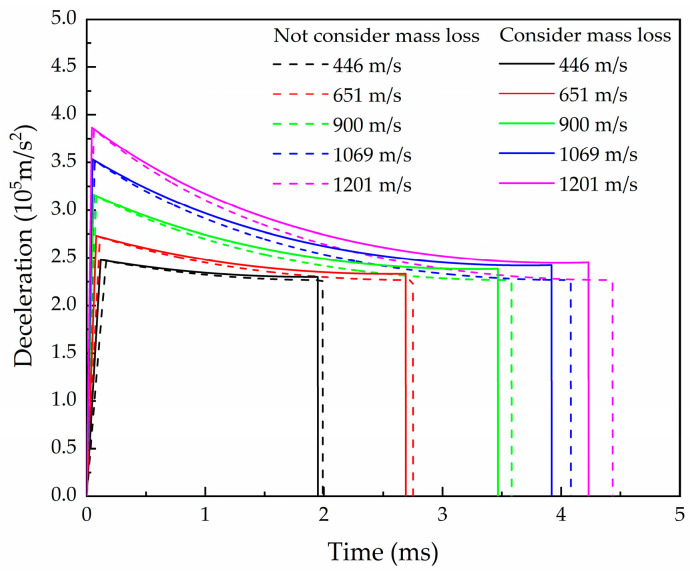
The relationship between deceleration and time considering mass loss.

**Figure 8 materials-16-03604-f008:**
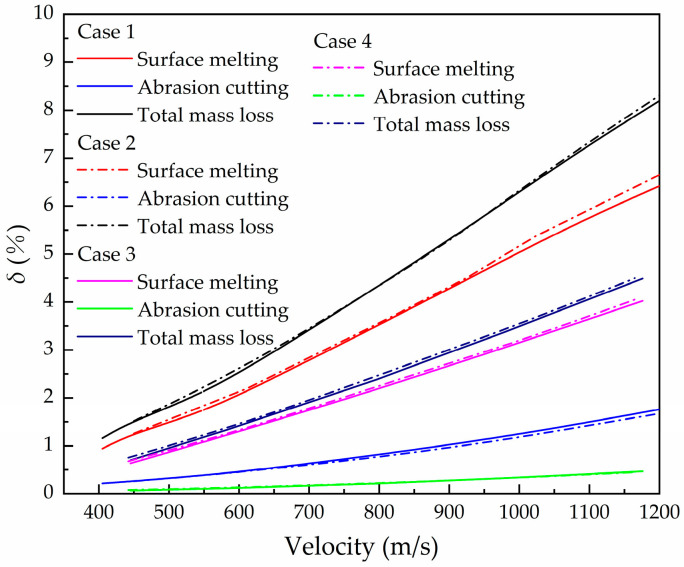
Relationship between mass loss of two mechanisms and initial penetration velocity.

**Figure 9 materials-16-03604-f009:**
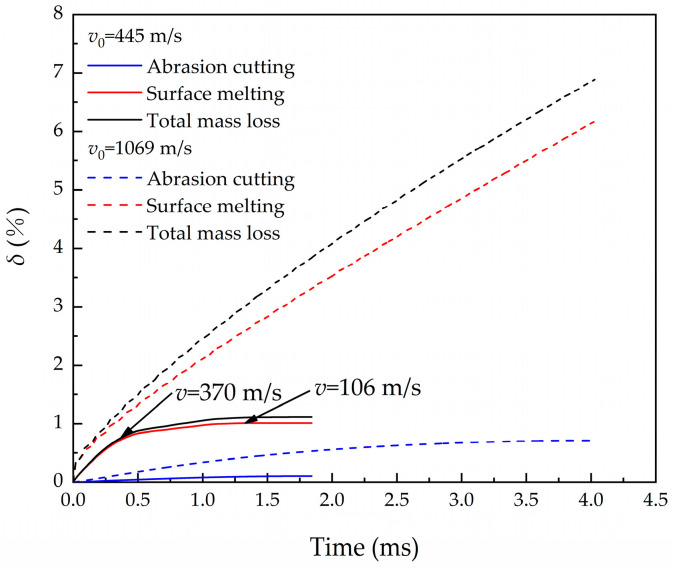
Relationship between mass erosion and time at two different initial velocities (445 m/s; 1069 m/s).

**Figure 10 materials-16-03604-f010:**
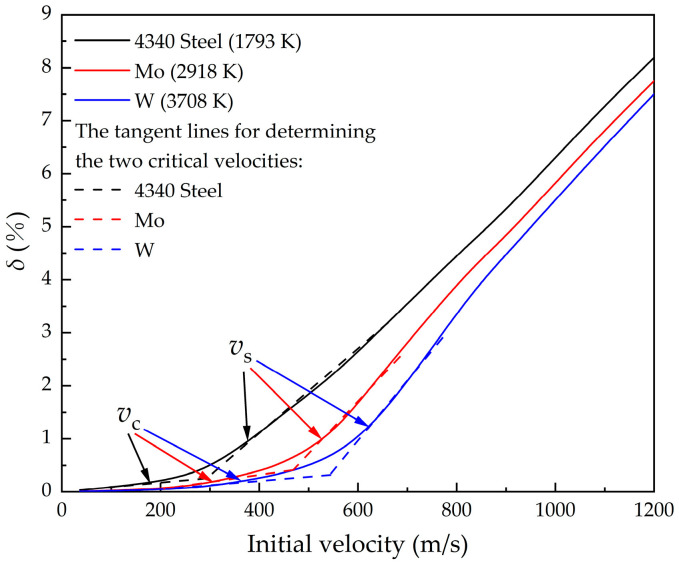
Effect of metals with different melting temperatures on mass erosion.

**Figure 11 materials-16-03604-f011:**
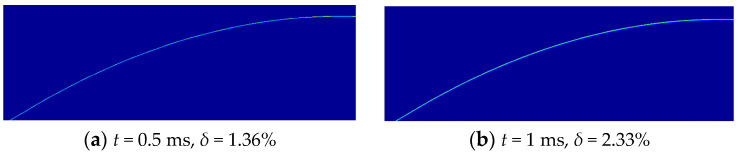
Temperature field of the nose surface (Case 1, *v*_0_ = 1069 m/s). (**a**–**h**) each picture shows the outer contour of the projectile nose after erosion at current time, and displays the temperature gradient of the nose cross-section. Where *δ* represents the percentage of mass loss at the current time. Due to the axisymmetric structure of the projectile, only half of the projectile nose is displayed.

**Figure 12 materials-16-03604-f012:**
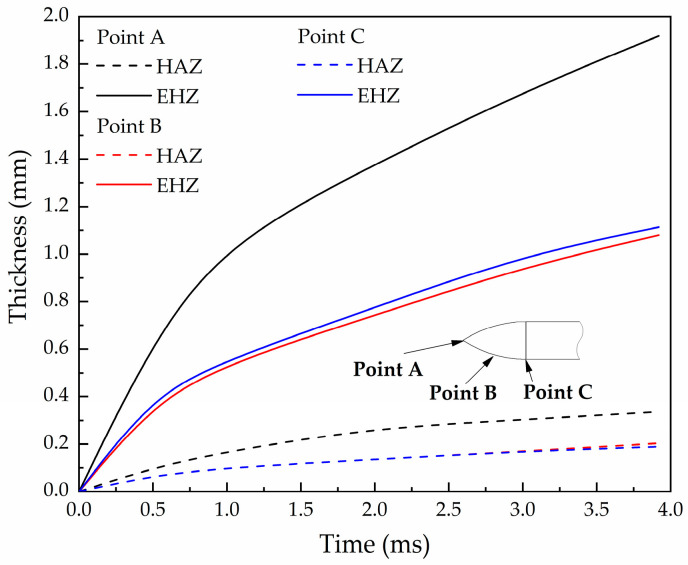
Thickness of the HAZ and EHZ on nose surface (Case 1, *v*_0_ = 1069 m/s).

**Table 1 materials-16-03604-t001:** Parameters of projectiles (4340 Steel).

	Case 1	Case 2	Case 3	Case 4
*m*_0_ (kg)	1.6	0.478	1.62	0.478
*d* (mm)	30.5	20.3	30.5	20.3
CRH	3	3	3	3
*L*_*n*_ (mm)	50.5	33.7	50.5	33.7
*L*_*p*_ (mm)	304.8	203.2	304.8	203.2

**Table 2 materials-16-03604-t002:** Parameters of concrete.

	Case 1	Case 2	Case 3	Case 4
Aggregate Material	Quartz	Quartz	Limestone	Limestone
*f*_c_ (MPa)	51	62.8	58.4	58.4
*ρ*_c_ (kg/m^3^)	2300	2300	2320	2320
Moh’s hardness of the aggregate	7	7	3	3
*K* _1_	2 × 10^−4^	2 × 10^−4^	6 × 10^−5^	6 × 10^−5^

**Table 3 materials-16-03604-t003:** Johnson-Cook constitutive model of 4340 steel material.

*ρ*_*p*_(kg/m^3^)	*A*	*B*	*C*	*n*	*m*	*T*_0_(K)	*T*_m_(K)
7850	792	510	0.014	0.26	1.03	298	1793

**Table 4 materials-16-03604-t004:** Parameters required for heat conduction and temperature calculation.

*k*_*p*_(W/(m·K))	*c*_*p*_(J/kg·K)	*k*_*c*_(W/(m·K))	*c*_*c*_(J/kg·K)	*β*
44.5	477	1.65	880	0.9

## Data Availability

Not applicable.

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
