# Peer review of "Study on Mass Erosion and Surface Temperature during High-Speed Penetration of Concrete by Projectile Considering Heat Conduction and Thermal Softening"

_materials, 2023, doi:10.3390/ma16093604_

Round 1
Reviewer 1 Report
I must commend you on the quality of your research work, as it was evident in your paper. The methodology used was appropriate, and the data analysis was rigorous. The conclusions drawn from your findings were logical, and the implications of your research were well-articulated. The quality of your writing was excellent, and the paper was well-organized.
Accepted in present form.
Author Response
Dear Editors:
Thanks to the experts and editors for your recognition of the work in this article. The author will conduct further research in this field in subsequent work, including using finite element programs to calculate mass erosion through secondary development, continuing to optimize calculation methods, and conducting experimental research on penetration and erosion of different sizes of projectiles and various forms of concrete. Wish you all the best!
Yours sincerely,
Reviewer 2 Report
The article Study on mass erosion and temperature during high-speed penetration of concrete by projectile considering heat conduction and thermal softening was considered. In this paper, some theoretical description of the model of an object in contact with an obstacle at speed is given. The model describes the temperature processes and the mass loss of the object. One of the important primary aspects of solving this problem is the thermoplastic contact of two bodies. The authors have done a great scientific work. The article may be published. But there are some questions and remarks in the article.
1. At the end of the first chapter, the authors need to give a detailed plan for the chapters. The authors need to briefly outline the goals of the research.
2. Studies were apparently carried out on the basis of formulas presented in the literature, are there any comparative full-scale experiments? What is their convergence?
3. Have numerical experiments been carried out in mathematical packages? Which ones?
4. For what reasons did the authors choose the material in the form of concrete? Concrete is an abstract material with a wide range of properties. How do the authors take into account the properties of this material in theory?
5. When describing the penetration of a given object under study into another object, the description of contact conditions apparently plays a huge role. How would the authors describe them?
6. The authors in the title of the article indicated the term thermal softening. It remains unclear what the term “thermal softening” means. Is there a plastic flow of metal?
7. How can the authors describe the distribution of heat in the object under study at certain points in time?
8. Can the authors present any calculation results?
9. An object moving at high speed is deformed upon impact with an obstacle, that is, it experiences plastic deformation, which reduces its speed, but leads to the release of heat. How do the authors take into account the moment of plastic deformation and how effective or ineffective is it in modeling?
10. Figure 11 is poorly understood for analysis. Are these the same pictures?
11. Under what conditions is the data object under study in Figure 12 correct? Logically, the object is deformed at the initial and subsequent points in time
12. At what minimum characteristics of concrete or speed of movement or other parameters does the object under study not deform and minimize mass loss?
Author Response
Dear Editors:
Thank you to the editors and reviewers for your reviews and suggestions, making the article more rigorous and reliable. After discussing with other authors, the main modifications made to the article and the answers to the reviewer are as follows:
- At the end of the first chapter, the authors need to give a detailed plan for the chapters. The authors need to briefly outline the goals of the research.
Answer: Thanks to the reviewer's suggestion. We have made modifications according to the reviewer's opinions and marked them in red font.
- Studies were apparently carried out on the basis of formulas presented in the literature, are there any comparative full-scale experiments? What is their convergence?
Answer: This study utilizes the theoretical formulas provided by the previous researchers, and the authors have made reasonable revisions to the formulas. For example, Eqs. (15) was modified to take into account the thermal softening of metals. At the same time, the effect of heat conduction during mass erosion was calculated. In previous studies, only surface temperature calculations were found in the literature of Guo [14] and Li [17], but this article considered more comprehensively and conducted coupling calculations.
Many experiments have been conducted on the process of mass erosion. This article cites the classic literatures [6,7] on mass erosion for comparison and is also a reference object for many model validation. However, the results of surface temperature only exist in theory, and no experimental research on surface temperature has been found. Using mass erosion for analysis is the only available method currently.
- Have numerical experiments been carried out in mathematical packages? Which ones?
Answer: The finite element programs, including LS-Dyna and Autodyn for calculation, is a commonly used method in the penetration process. Although the phenomenon of mass erosion was discovered earlier, but scholars only conducted calculations through experiments and theory, the numerical research on mass erosion has only developed for more than ten years. The implementation of mass erosion through finite element programs has not yet been effectively achieved. Coupling the finite element software and the program generated by the mathematical model through an interface for calculation is also our next work plan.
- For what reasons did the authors choose the material in the form of concrete? Concrete is an abstract material with a wide range of properties. How do the authors take into account the properties of this material in theory?
Answer: There are many types of concrete, and the concrete considered in this article is a common material used in engineering protective structures, usually made by mixing and curing cement, mortar, and aggregates. Therefore, it is also a common form of concrete when projectile penetration occurs in engineering. Its density and compressive strength are important physical and mechanical parameters of concrete, and these two parameters are also important core indicators that affect the penetration process, Therefore, in theoretical calculations, this parameter is used to express the resistance and surface pressure (Eq. (1)) of the projectile nose.
- When describing the penetration of a given object under study into another object, the description of contact conditions apparently plays a huge role. How would the authors describe them?
Answer: In the theoretical calculation of projectiles, the contact condition during the interaction between projectiles and targets is mainly the resistance of the projectiles (Equ. 6), and the resistance term of the projectiles is mainly derived from the cavity expansion theory. The formula is composed of and as variables, which is derived from the surface pressure (Eq. (1)) and integrated by considering the nose shape to obtain the resistance formula. This resistance formula has been widely verified in numerical calculations.
- The authors in the title of the article indicated the term thermal softening. It remains unclear what the term “thermal softening” means. Is there a plastic flow of metal?
Answer: Thermal softening refers to the mechanical behavior of metal materials that exhibit softening at high temperatures, which is an inherent property of the material. During mechanical testing, the tensile strengths obtained at different environmental temperatures are different, especially under high-temperature conditions where the strength of this metal sharply weakens and exhibits softening characteristics. During the penetration process, the surface temperature of the projectile can reach thousands of degrees Kelvin, so considering thermal softening in the calculation of mass erosion can make the model more reasonable.
- How can the authors describe the distribution of heat in the object under study at certain points in time?
Answer: There are two ways. First, we can use the temperature field as shown in Figure 11. Temperature is described in the form of heat. This paper shows the temperature field at different time points. Because the thickness of the high-temperature zone is very small during the penetration process due to the extremely short penetration time, we give the relationship between the thickness of the high-temperature Heat-affected zone and the thickness of the whole heat conduction zone at three typically different points. In this way, the readers can interpret the two forms of expression according to their requirements.
- Can the authors present any calculation results?
Answer: Figure 6 shows the calculation results of penetration depth and mass erosion, which are compared with experimental results, indicating the effectiveness of the model and calculation method. An extension for analysis was shown after the verification. Figure 8 and Figure 9 show the analysis of the proportion of melting erosion and cutting erosion and discuss the influence of speed on the proportion of the two erosion. Figure 11 and Figure 12 show the temperature calculation results with corresponding analyses.
- An object moving at high speed is deformed upon impact with an obstacle, that is, it experiences plastic deformation, which reduces its speed, but leads to the release of heat. How do the authors take into account the moment of plastic deformation and how effective or ineffective is it in modeling?
Answer: In theoretical calculations, the influence of plastic deformation is mainly considered through iterative calculations using the application of surface pressure as a boundary condition. The surface pressure is given by Eq. (1). The correctness of surface pressure is essentially verified by the results of penetration depth, so it is reliable. In the model establishment of this article, plastic deformation has a certain influence on temperature. The plastic deformation work converted to heat energy is calculated using Eq. (10), and the total temperature rise of the projectile is accumulated.
- Figure 11 is poorly understood for analysis. Are these the same pictures?
Answer: These images are temperature fields at different times during the penetration process, and there are obvious differences between these images. One is the size of the outer contour, which changes due to the mass loss. The other is the thickness layer in the high-temperature zone, which is calculated from heat generation and heat conduction. This article couples mass loss and temperature so can be expressed in one image. Since the thickness layer accounts for a small proportion of the whole nose, in order to describe the thickness in detail, the relationship between the thickness of the high-temperature heat affected zone and the thickness of the entire heat conduction zone at three typical locations (nose tip, nose middle, and nose end) is given as shown in Figure 12.
- Under what conditions is the data object under study in Figure 12 correct? Logically, the object is deformed at the initial and subsequent points in time
Answer: The obtain of Figure 12 is conditional, that is, the calculation corresponding to the experiment in reference [6,7] has been explained, that is also corresponding to the projectile and concrete of Case1 in Table 1, and penetrating at an initial velocity of 1069m/s. The analysis presented here as a typical case is based on the parameter analysis after the model validation conducted through numerical calculations in this paper.
- At what minimum characteristics of concrete or speed of movement or other parameters does the object under study not deform and minimize mass loss?
Answer: Within the scope of high-speed research in this article, the mass loss is relatively significant, therefore it has valuable to research. As long as the projectile occurs penetration behavior, there is mass loss, as shown in Figure 10. However, the lower the velocity, the mass loss is closer to zero. The value of the mass loss is related to the shape, velocity, and concrete parameters of the projectile, and calculations and analysis are needed for the specific problems. However, from experience, it is generally not obvious that erosion occurs during low-speed penetration, At this point, mass loss can be a secondary influencing factor and can be ignored. In numerical calculations, the projectile was often considered as a rigid body with initial constant mass for the model established.
Thank you again for your valuable suggestions, as well as the editorial department. Wish you all the best!
Yours sincerely,
Reviewer 3 Report
The article presents a proposal to evaluate mass erosion and temperature during high-speed penetration of concrete by projectile considering heat conduction and thermal softening. Although the proposed theme presents an originality of research, which is encouraging for publication, many points of the article are confusing. Please see the comments below:
- The title and the abstract are flawed because they do not make it clear to readers what are the main research points and what are the knowledge gaps that are being studied. It is important that after reading the title and abstract it is very clear to the readers of the article what is being investigated. Please review this.
- The intro is flawed. The authors used many generic references. Instead, use references that add to what is being discussed in your article. See for example the first paragraph of the article. The authors use 10 references to present information that is widely known to authors in the field. Review this.
- Include which knowledge gaps are being explored in your research. In the introduction, include a paragraph that expresses the main innovation points of your research.
- Section 2: present the limitations of the presented equations. Explain when the information in this section does not apply.
- Section 4: a critical point of the article is that the authors do not compare the results obtained with similar research. For your article to have scientific validity, it is extremely important to compare the results obtained with other similar research.
- Section 4.2: it is extremely important to highlight the differences observed in Figure 11 and compare the results obtained with other studies. Authors who do not have a level of understanding of the research carried out by the authors will not understand exactly what is being discussed in this figure.
- The conclusions need to summarize the main points discussed in the research. In the current format the conclusions are generic and are flawed. Review conclusions after responding to previous comments.
Not applicable
Author Response
Dear Editors:
Thank you to the editors and reviewers for your reviews and suggestions, making the article more rigorous and reliable. After discussing with other authors, the main modifications made to the article and the answers to the reviewer are as follows:
- The title and the abstract are flawed because they do not make it clear to readers what are the main research points and what are the knowledge gaps that are being studied. It is important that after reading the title and abstract it is very clear to the readers of the article what is being investigated. Please review this.
Answer: The abstract and title have been optimized, highlighting the erosion calculation considering thermal softening and surface temperature, as well as the scientific temperature calculation method using the assumption of a non-adiabatic process and considering heat conduction. This is corresponding to 'considering heat conduction and thermal softening' in the title.
- The intro is flawed. The authors used many generic references. Instead, use references that add to what is being discussed in your article. See for example the first paragraph of the article. The authors use 10 references to present information that is widely known to authors in the field. Review this.
Answer: The literature in the first paragraph has been partially deleted and only necessary references have been retained.
- Include which knowledge gaps are being explored in your research. In the introduction, include a paragraph that expresses the main innovation points of your research.
Answer: The sixth paragraph in the introduction has been optimized, mainly highlighting the shortcomings of thermal softening not being considered by existing calculation models, as well as the assumptions used in temperature calculation of existing models (especially adiabatic assumptions). In the last paragraph of the introduction, the improvement of the model in this article was also expressed, corresponding to the abstract either.
- Section 2: present the limitations of the presented equations. Explain when the information in this section does not apply.
Answer: The equations established in this article have been verified by experimental results, and their applicability is relatively broad. From the comparison and verification with experimental results later, it can be seen that the projectile velocity in this article is applicable between 400~1201m/s, which is within the range of "high speed" in the title. In addition, based on its regularity, it can be seen that it also conforms to the erosion law when it is below 400m/s, but when it is above 1201m/s, due to the possible fragmentation phenomenon of the projectile, the mass loss at this time is not a simple erosion process, so it needs to be reconsidered. The speed limitations have been explained in the conclusion.
- Section 4: a critical point of the article is that the authors do not compare the results obtained with similar research. For your article to have scientific validity, it is extremely important to compare the results obtained with other similar research.
Answer: Thanks for the valuable opinions put forward by the expert, this paper has established a coupled calculation model of mass erosion and heat conduction. And calculated the proportion of mass erosion, cutting erosion and melting erosion, and the temperature distribution on the nose surface. The mass erosion has been compared with the test results, as shown in Figure 6, which shows that the model is correct and scientific. The subsequent analysis is a derivative calculation based on this model, and similar research on erosion proportion and temperature distribution has not been found in other literature.
- Section 4.2: it is extremely important to highlight the differences observed in Figure 11 and compare the results obtained with other studies. Authors who do not have a level of understanding of the research carried out by the authors will not understand exactly what is being discussed in this figure.
Answer: Figure 11 is the temperature field calculated based on the model in this article. Due to the adiabatic assumption, and the heat conduction was not considered in previous literature, the surface temperature obtained is only a single value rather than a field with a certain thickness. This figure is the first obtained in current research, and any similar conclusion has not been found in previous literature. The surface temperature and Guo's calculation results are consistent, as explained in Section 4.2. Due to the small thickness of the high-temperature layer, for ease of understanding, Figure 12 has been added, which shows the thickness curves of HAZ and EHZ at three typical positions.
- The conclusions need to summarize the main points discussed in the research. In the current format the conclusions are generic and are flawed. Review conclusions after responding to previous comments
Answer: Thank you for the valuable opinions provided by the expert. It should be explained that among the three conclusions, only the first one regarding "The proposal of cutting and melting is closely related to the hardness of aggregate in concrete" has been obtained through experiments. However, it is included in the conclusion of this article because it has been verified through theoretical calculations by the proposed model, And this conclusion will help to understand Conclusion (2) and Conclusion (3) below. Conclusion (2) and Conclusion (3) are both obtained through model calculations. The validation of the model is detailed and described in Section 4.1.1, so it is believed that the conclusions are reliable. The main content of these conclusions is the analysis of the proportion of melting erosion and cutting erosion and their relationship with projectile velocity. These conclusions have not been reported in previous literature, and are helpful for the optimization of projectile design.
Thank you again for your valuable suggestions, as well as the editorial department. Wish you all the best!
Yours sincerely,
Round 2
Reviewer 3 Report
The authors responded appropriately to previous review comments.